# Intra-Articular Surgical Reconstruction of a Canine Cranial Cruciate Ligament Using an Ultra-High-Molecular-Weight Polyethylene Ligament: Case Report with Six-Month Clinical Outcome

**DOI:** 10.3390/vetsci11080334

**Published:** 2024-07-25

**Authors:** Sven Ödman, Antonin Martenne-Duplan, Marlène Finck, Antonin Crumière, Bastien Goin, Philippe Buttin, Eric Viguier, Thibaut Cachon, Krister Julinder

**Affiliations:** 1Animal ArtroClinic i Söderköping AB, Ringvägen 40, 614 33 Söderköping, Sweden; 2Centre Hospitalier Vétérinaire Massilia, Animedis, IVC Evidensia France, 13012 Marseille, France; 3Novetech Surgery, 13bis Boulevard Tzarewitch, 06000 Nice, France; a.crumiere@novetech-surgery.com (A.C.); b.goin@novetech-surgery.com (B.G.); 4VetAgro Sup, Interactions Cellules Environnement (ICE), University of Lyon, 69280 Marcy l’Etoile, France; 5Univ Lyon, Univ Gustave Eiffel, Univ Claude Bernard Lyon 1, LBMC UMR T_9406, 69675 Bron Cedex, France; 6Independent Researcher, 74370 Villaz, France

**Keywords:** arthroscopy, cranial cruciate ligament, dog, synthetic ligament reconstruction, UHMWPE

## Abstract

**Simple Summary:**

Rupture of the cranial cruciate ligament (CrCL) is a common condition in dogs, causing instability of the stifle, pain, lameness, osteoarthritis, and meniscal tears. Common treatments consist of tibial osteotomies or extra-articular stabilization of the stifle according to the size and weight of the dog. The most marginal routine practice is intra-articular stabilization, which consists of replacing the ruptured CrCL with an organic graft or a synthetic implant. This case report describes the technique used for the intra-articular synthetic reconstruction of the CrCL under arthroscopic guidance in a dog. This report is supported by a six-month clinical outcome assessment consisting of orthopedic examinations and radiographs. The dog showed quick resumption to normal gait and standing posture with no worsening of osteoarthritis and synovial effusion or complications. Mild signs of stifle instability were observed with no impact on clinical outcome. This technique could be considered an alternative for the treatment of CrCL rupture in dogs, but it needs confirmation from additional clinical studies with more dogs.

**Abstract:**

The intra-articular reconstruction of the cranial cruciate ligament (CrCL) by an organic graft or a synthetic implant allows the restoration of physiological stifle stability. This treatment is still marginal in routine practice. A Rottweiler presented an acute complete CrCL rupture treated using an ultra-high-molecular-weight polyethylene (UHMWPE) implant. The latter was positioned under arthroscopic guidance and fixed with interference screws through femoral and tibial bone tunnels. The dog was weight-bearing just after surgery and resumed normal standing posture and gait after one month, with mild signs of pain upon stifle manipulation. At three months postoperatively, minimal muscle atrophy and minimal craniocaudal translation were noted on the operated hindlimb, with no effects on the clinical outcome. The stifle was painless. At six months postoperatively, standing posture and gait were normal, muscle atrophy had decreased, the stifle was painless, and the craniocaudal translation was stable. On radiographs, congruent articular surfaces were observed without worsening of osteoarthrosis over the follow-up, as well as stable moderate joint effusion. Replacement of a ruptured CrCL with a UHMWPE ligament yielded good functional clinical outcome at six months postoperatively. This technique could be considered an alternative for the treatment of CrCL rupture in large dogs, but it needs confirmation from a prospective study with more dogs.

## 1. Introduction

Cranial cruciate ligament (CrCL) rupture affects up to 4.87% of dogs [1] and is the most common stifle orthopedic injury requiring veterinary care [2]. Middle-to-large-breed, old, and overweight dogs are predisposed to this pathology [1,3,4]. The CrCL prevents craniocaudal instability, internal rotation, and hyperextension of the tibia relative to the femur [5]. Its rupture causes pain, lameness, initiation or aggravation of osteoarthritis (OA), and meniscal tears [5]. In large dogs (>15 kg), surgical stabilization is often used to treat the instability [6,7], thus limiting OA and meniscal injuries [5].

Proximal tibial osteotomy modifies the biomechanical conformation of the stifle and suppresses the need for a functional CrCL during weight-bearing. It results in craniocaudal stabilization of the joint during locomotion [8]. The main techniques used are tibial-plateau-leveling osteotomy (TPLO) [7,9] and tibial tuberosity advancement (TTA) [7,10]. Less common techniques include Cora-based leveling osteotomy (CBLO) [11], tibial tuberosity osteotomy (TTO) [12], and cranial closing wedge osteotomy (CCWO) [13]. 

Extra- or intra-articular stabilization of the stifle using sutures or implants is less invasive than tibial osteotomy techniques. Extra-articular stabilization includes lateral fabellotibial sutures [14] performed with various materials such as polyester, high-molecular-weight polyethylene or nylon thread secured with crimps [15,16], and ultra-high-molecular-weight polyethylene (UHMWPE) sutures positioned outside of the joint like the TightRope [17] and the RUBY implant [18]. Tibial osteotomies are preferable to extra-articular sutures in large dogs [19], mainly due to their weight, which exerts a greater force on implants [20]. 

Unlike tibial osteotomy and extra-articular stabilization, intra-articular stabilization aims at solving the direct cause of the instability by reconstructing the disrupted CrCL and restoring its function [21]. It is mainly used in large breed dogs with autografts [21,22], allografts [23,24,25], and synthetic implants [26,27,28,29]. 

Due to new polymers and implants, arthroscopic implantation of synthetic implants is being increasingly used to treat human anterior cruciate ligament rupture [30]. Although intra-articular reconstruction of the CrCL has also received growing interest in veterinary research [26,27,28,29,31,32], it is still marginal in routine practice [6,7]. A UHMWPE implant fixed by interference screws (ISs) has shown good ex vivo biomechanical results on canine stifles [33,34,35,36] as well as satisfying clinical outcomes for the reconstruction of canine caudal cruciate ligament [37] and the treatment of multiligament stifle injuries in dogs and cats [38].

This article reports the use of this implant for the arthroscopic reconstruction of a CrCL rupture in a dog and its six-month clinical outcome. 

## 2. Case Presentation

### 2.1. Clinical History

A 47 kg (body score condition: 2/5 [39]), three-year nine-month-old entire male Rottweiler used as a companion dog was presented with a one-month history of lameness and abnormal standing posture of the right hindlimb. The symptoms had started suddenly after playing.

### 2.2. General and Orthopedic Examination

General examination was unremarkable. Weight-bearing lameness of the right pelvic limb was scored at 1/4 for both walking and trotting [27]. In standing posture, the dog was intermittently weight-bearing on the right hindlimb. Measurement of thigh muscle mass performed at 50% of thigh length was comparable to the contralateral one [40]. A positive direct drawer test showed a 5 mm craniocaudal movement of the tibia relative to the femur, and the indirect cranial drawer was positive. The pivot shift was moderate yet present. Manipulation of the stifle joint in flexion, extension, and internal rotation was painful (pain scored 1/3 [41]). No crepitus nor meniscal click were detected. These findings were consistent with CrCL rupture.

### 2.3. Diagnostic Imaging under Anesthesia

Under anesthesia, compression stress radiographs (MULTIX Impact, Siemens Healthineers, Forchheim, Germany) revealed cranial displacement of the tibial plateau relative to the femoral condyles (Figure 1). The tibial plateau angle was measured at 25.8°. There was no radiographic evidence of bone fracture. The presence of enthesophytes at the patellar apex and on the tibial condyles was observed (Figure 1). Markedly increased intra-articular soft tissue opacity leading to cranial compression of the intra-articular fat pad was reported, with distension of the caudal synovial joint capsule consistent with moderate joint effusion (Figure 1). These features supported the findings from the orthopedic examination. An intra-articular synthetic reconstruction of the CrCL under arthroscopic guidance was decided. 

### 2.4. Surgical Treatment

A UHMWPE implant (Novalig 8000, Novetech Surgery, Nice, France), with an ultimate load of 8000 N adapted to the size and weight of the dog and fixed by two 6 × 25 mm ISs suitable for the implant and the dog’s bone size, was used to repair the CrCL according to the manufacturer’s recommendations (Figure 2). These choices determined the size of the bone tunnels for fixation.

Under anesthesia, the dog was positioned in dorsal recumbency with the right stifle placed on a support and locked in full flexion using a rubber band passing over the talus to maintain the joint open and facilitate access to the intra-articular compartment (Figure 3A and Figure 4A). The hindlimb was prepared aseptically and draped for surgery. 

A first 6.0 mm parapatellar craniolateral skin incision was made at the femorotibial junction to introduce the arthroscope (30° 4.0 × 156 mm, Arthrex, Naples, FL, USA) with a 4.5 mm cannula (Arthrex, Naples, FL, USA) (Figure 3A). A second 6.0 mm parapatellar craniomedial skin incision was made at the femorotibial junction to introduce the shaver (4.0-mm Torpedo Shaver, Arthrex, Naples, FL, USA) (Figure 3A), which was used to remove the fat pad. Arthroscopic evaluation of the joint revealed a minor tear of the surface of the medial meniscus (type-4 lesion [42]) and a completely ruptured CrCL (Figure 3B). Both the remnants of the CrCL and the damaged tissue of the meniscus were removed with the shaver. 

The shaver was then removed to introduce the hook of the drilling guide through the craniomedial incision. The hook of the guide was anchored on the tibial plateau caudally to the transverse ligament of the menisci at the tibial insertion of the CrCL footprint under arthroscopic guidance (Figure 4A,B) [43]. A third 10.0 mm skin incision was made medially on the proximal tibia to anchor the sleeve of the aiming device on the tibia (Figure 4A) as caudal as possible in order to drill the tunnel into a high bone density region for stronger IS fixation. The sleeve was anchored on the proximo-distal axis of the tibia to drill a tunnel that fitted the size of the IS in order to have it fully inserted into the bone tunnel. The position and orientation of the tibial tunnel were secured using a 2.5 mm Kirschner wire introduced into the sleeve of the aiming device (i.e., into the medial incision on the proximal tibia) until the hook. Then, the aiming device was removed, maintaining the Kirschner wire in position. The tibial bone tunnel was drilled outside-in over the Kirschner guide wire using a 5.0 mm cannulated drill bit (Figure 4B). 

After removing the Kirschner guide wire and the cannulated drill bit, the femoral insertion footprint of the CrCL was located using the arthroscope. The position and orientation of the femoral tunnel were secured using a 2.5 mm Kirschner wire introduced into the parapatellar craniomedial skin incision. The position was determined by targeting the center of the CrCL footprint, and the orientation was determined to provide a tunnel that fitted the size of the IS in order to have it fully inserted into the bone tunnel, similarly to the tibial tunnel. No aiming device was used to drill the femoral tunnel. The tunnel was drilled inside-out over the Kirschner guide wire using a 5.0 mm cannulated drill bit (Figure 4B). 

A fourth 10.0 mm incision was made on the lateral aspect of the femur to provide access to the lateral entrance of the femoral tunnel. The extremity of the cannulated drill bit was retrieved. This latter was gently pulled and maintained inside the femoral tunnel to leave the intra-articular entrance of the femoral tunnel free.

Using the shaver through the craniomedial skin incision, the femoral tunnel was cleaned, and its entrance inside the joint was cleaned and smoothened. Then, the shaver was introduced from distal to proximal into the tibial tunnel up to the intra-articular space to clean the tunnel and smooth the entrance within the joint. 

Both tunnels were compacted using the 6 × 25 mm ISs intended for implant fixation. The tibial tunnel was compacted through the medial incision on the proximal tibia, and the femoral tunnel was compacted through the parapatellar craniomedial incision. 

The cannulated drill bit in the femoral tunnel was pushed to have its extremity exiting through the parapatellar craniomedial skin incision. The puller wire section of the implant (Figure 2) was placed into a hole present in the drill bit, which was still positioned in the femoral tunnel to act as a passing loop. The implant was introduced into the parapatellar craniomedial skin incision and passed from distal to proximal through the femoral tunnel by pulling the drill bit with the implant. The extremity of the implant was retrieved laterally. 

A cerclage wire loop was introduced in the tibia tunnel through the medial incision on the proximal tibia. The wire loop was retrieved with forceps to be pulled out of the joint at the parapatellar craniomedial entrance. The second puller wire section of the implant was then passed in the loop using the forceps and was pulled from proximal to distal through the tibial tunnel and retrieved medially (Figure 2 and Figure 4C). A 6 × 25 mm cannulated titanium IS (Novetech Surgery, Nice, France) (Figure 2c) mounted on a Kirschner wire guide and on a cannulated ratchet screwdriver (Novetech Surgery, Nice, France) was inserted into the parapatellar craniomedial incision to secure the implant from inside-out in the femoral tunnel (Figure 3D and Figure 4C,D). The Kirschner wire guide and arthroscopic guidance ensured that the IS insertion into the tunnel followed the drilling axis. The head of the IS was flushed with the articular surface (Figure 3D). The rubber band maintaining the limb in full flexion was then removed to let it hang free and allow appropriate tensioning, which was adjusted by rolling the implant around a Kocher forceps to apply sufficient force (Figure 4D). A second 6 × 25 mm IS (Novetech Surgery, Nice, France) (Figure 2), also mounted on a Kirschner wire guide and on a cannulated ratchet screwdriver to control insertion, was placed outside-in in the tibial tunnel using the medial incision on the proximal tibia to lock the implant (Figure 4E,F). The extremity of the tibial IS was not flushed with the articular surface. The extremities of the implant were cut at the entrance of both tunnels (Figure 4E,F). Plane-by-plane closure was then performed.

### 2.5. Immediate Postoperative Evaluation

The immediate postoperative evaluation showed that the stifle had no residual drawer sign in the craniocaudal plane and no pivot shift. Compression stress radiographs (Mobilett Mira Max, Siemens Healthineers, Forchheim, Germany) showed appropriate positioning of the tunnels and satisfactory implantation of the ISs fully embedded inside the tunnels along the drilling axes, with congruent articular surfaces (Figure 5).

### 2.6. Postoperative Management

The dog received cefalexin (10–25 mg/kg p. o. q8-12h) (Kefavet^®^ vet., Orion Pharma Animal Health, Danderyd, Sweden) for six days, tramadol (2.5 mg/kg p. o. q8h) (Tramadol 2care4, 2care4 Generics, Esbjerg, Denmark) for six days, and nonsteroidal anti-inflammatory drug (firocoxib, 5 mg/kg p. o. q24h) (Previcox, Boehringer Ingelheim Animal Health, Copenhagen, Denmark) for 14 days as postoperative medication. Full rest was recommended during the first two weeks. The dog was then prescribed a four-month physiotherapy rehabilitation program, which included stretching and manipulating the operated hindlimb, weight-bearing exercises, and leash walking progressively increasing in length, duration, and intensity. The initial recommendation was to release it from physiotherapy but still walk it on a leash after four months and to let it resume normal activity without any restrictions after six months. In the end, the owner reported that the program was followed only for three months, and the dog was allowed to resume normal activity since it seemed to be doing well. 

## 3. Results

Postoperative control visits consisted of clinical and orthopedic examinations after one, three, and six months, including three- and six-month postoperative radiographs. At one month postoperatively, no lameness was observed at both walking and trotting (both scored 0/4 [27]). The dog showed discomfort in the standing position with occasional lifting of the paw and reacted aggressively upon manipulation of the stifle, which indicated a 1/3 pain score [41]. The stifle seemed stable, but because of the dog’s reaction, the sensitivity of the test was limited. No general anesthesia was induced, and no additional drawer test or manipulation of the stifle was performed to avoid excessive stress on the animal. No swelling was observed. The owner reported that the dog resumed a normal standing position two months after surgery.

At three months postoperatively, standing posture and gait at both walking and trotting were normal (both scored 0/4 [27]). A slight 1 cm muscle atrophy was noted on the operated hindlimb compared to the contralateral one. Manipulation of the joint in flexion, extension, and internal rotation of the tibia was similar to that of the contralateral limb and caused no reaction in the dog. This suggested an absence of pain (scored 0/3 [41]). No crepitus nor meniscal click were observed. The direct cranial drawer test performed under general anesthesia revealed a slight (3 mm) craniocaudal movement of the tibia relative to the femur. The indirect cranial drawer test showed no instability. Compression stress radiographs showed satisfactory congruent articular surfaces (Figure 6). The ISs were aligned along the tunnel axes without any deformation of the latter when compared to immediate postoperative control radiographs (Figure 5 and Figure 6). Closure of both femoral and tibial tunnels was observed, indicating bone healing (Figure 6). There was no radiographic worsening of the OA (Figure 6). Synovial effusion remained moderate (Figure 6). The owner reported that the dog had resumed normal activity levels four months after surgery.

At six months post-surgery, standing posture was normal with normal gait at both walking and trotting (lameness scored 0/4 [27]; Appendix A). The slight muscle atrophy had reduced to a 0.5 cm difference compared to the contralateral hindlimb. Manipulation of the joint in flexion, extension, and internal rotation of the tibia was similar to that of the contralateral joint and caused no reaction in the dog, suggesting the absence of pain (scored 0/3 [41]). No crepitus nor meniscal click were observed. The direct cranial drawer test performed under general anesthesia indicated a slight (3 mm) craniocaudal translation. The indirect cranial drawer test was negative. Compression stress radiographs showed satisfactory congruent articular surfaces without worsening of OA (Figure 7). Synovial effusion was still moderate (Figure 7). When compared to the three-month postoperative radiographs, no movement of the ISs along the tunnel axes nor deformation of the tunnels was observed, and closure of both femoral and tibial tunnels continued (Figure 6 and Figure 7). No signs of postoperative complications were observed during follow-up.

## 4. Discussion

A CrCL rupture in a large-breed dog was treated by synthetic intra-articular reconstruction using a UHMWPE implant fixed by ISs in femoral and tibial tunnels. The functional clinical outcome at six months postoperatively indicated (i) painless functional stifle with normal gait at both walking and trotting, (ii) mild craniocaudal translation observed only with direct drawer test, and (iii) no signs of infection or changes in OA and joint effusion. The satisfactory joint function of the operated stifle was similar to that of the contralateral hindlimb and comparable to the best outcomes obtained in previous intra-articular CrCL reconstructions performed with grafts [24,25,44,45], synthetic implants made of different materials [27,28] or UHMWPE sutures [31,46]. 

Recurrence of mild craniocaudal joint translation without negative impact on gait has also been reported in previous CrCL reconstructions [24,28,44]. This non-pathological 3 mm craniocaudal translation, observed only with the direct drawer test, might be expected and was at the threshold value to be considered as null [24], suggesting relative joint stability. The absence of instability with the indirect drawer test, similar to weight-bearing after a TPLO [8,44], supported this satisfactory outcome. Fluoroscopy would have revealed the in vivo three-dimensional kinematics of the joint during activity to assess functionality and stability more objectively [47]. Several parameters like positioning [48], integrity [27,28], and sliding [49] of the implant may explain the recurrence of mild craniocaudal translation.

The implant aimed to reproduce the mechanical role of the ligament in the joint. Isometric implantation should help to avoid overstrain and micromovements of both the implant and the ISs to reduce the risk of slippage [49,50]. However, CrCL insertion footprints extend over the tibia and femur [43,51,52], and their accessibility is influenced by the position of the limb during surgery and by the morphology of the dog’s bones (i.e., tibial plateau slope and proximity of femoral condyles). This makes the identification and accessibility of the footprints’ centers for appropriate positioning of the tunnels more complex.

Arthroscopy helped to identify the correct locations to achieve physiological implantation. Maintaining the limb in full flexion made the tibial plateau more accessible and facilitated the hooking of the aiming device on the tibia. The outside-in drilling approach may have simplified the procedure as it avoided interference with the intra-articular space. 

The full flexion of the limb also facilitated the accessibility of the femoral intercondylar space. The inside-out approach helped to target the center of the femoral CrCL footprint on the lateral condyle with a guide wire [27,28]. However, since the drill bit is always larger than the guide wire, the position of the drill bit may have to be adjusted to preserve the integrity of the medial condyle during drilling, which may influence the position and/or angulation of the tunnel [24]. By using a measurement method applied to a drilling technique developed ex vivo [51,52], the optimal positioning of the femoral tunnel entrance should have been 4 mm more caudal.

An alternative for a more accurate femoral tunnel position would be to drill the tunnel outside-in to avoid any interference with the medial condyle [51,52] and insert an IS outside-in. Yet, this solution is limited since outside-in fixation provides weaker pull-out strength than inside-out fixation when using ISs [36]. This logic also applies to the outside-in fixation that was used in the tibial tunnel, which might represent a limitation of this implantation technique. 

The position of the implant in relation to the ISs inside the tunnels might also play a functional role in the stability of the CrCL reconstruction. Since the implant is pressed against a single side of the tunnel wall, it is not positioned and tensioned at the center of the footprints. The most relevant method might thus be to drill the tunnels a few millimeters away from the footprint center, which would help to secure the implant closer to that center. Another parameter is the position of the ISs inside the tunnels. In the present case, the tibial IS was inserted outside-in and was not screwed flush with the joint surface. Therefore, the working section of the implant was secured more distally inside the tibial tunnel. This may allow more freedom of movement and may explain why the craniocaudal translation was observable only with the direct drawer test. Inserting the ISs flush with the articular surface in the tibial tunnel would reduce the length of the working section of the implant, which might help limit implant relaxation and, thus, craniocaudal translation. Overall, correct tunnel positioning and drilling are challenging and influenced by the experience of the surgeon, who makes a subjective decision according to the dog’s morphology.

Implant strength and integrity are other major parameters in the functional outcome of the technique [27,28,53]. UHMWPE implants have suffered fewer breakages [31,46] than other synthetic models made of ultra-high-molecular-weight polyethylene terathalate and non-expanded polytetrafluoroethylene [27] or polyethylene terephthalate [28], and who were not recommended for clinical application in CrCL reconstruction. The UHMWPE implant used here has an ultimate load superior to the maximal strength of a physiological CrCL [35,54], and no rupture was reported in quasi-static pull-out tests [33] and cyclic loading tests [34]. Its rupture was unlikely to occur, even under cyclic or torsional fatigue hypothesized as a possible cause of implant rupture in previous synthetic reconstructions with other polymers [27,28]. An exception might occur with cyclic fatigue if the implant were to be progressively damaged by abrasive fixations owing to repeated micromovements during locomotion. While the implant was expected to resist the constraints generated by canine locomotion, arthroscopy would have allowed direct observation of its integrity over time [28].

With this UHMWPE implant intact, implant slippage at the bone/IS interface was the only failure mode observed during ex vivo biomechanical tests performed on intra-articular CrCL reconstruction [33,34,35,36,55]. Fixation by ISs showed maximal quasi-static pull-out strength before implant slippage at 690 ± 115 N [33] and was validated in fatigue tests up to trotting effort (i.e., 210 N) before slippage occurred [34]. These results remain below native CrCL strength, which can resist forces up to 888 N [33]. 

Fixation by ISs provides pull-out strength distributed along the bone tunnel [55] and avoids using a knot fixation that may slip with UHMWPE [56] but is dependent on the dog’s bone quality [57]. In the event of low bone stiffness due to low bone quality (i.e., insufficient bone quantity and/or bone density), the ISs may deviate from the tunnels’ axes during insertion or move from their initial position after implantation. This may decrease or unbalance the compressive forces exerted on the implant along the tunnels’ axes, inducing local overstrains on the ISs and the implant, which would then affect functional fixation.

Previous synthetic reconstructions used a four-tunnel approach with ISs and screws with spiked washers [27,28]. This fixation method aimed to resist a quick return to vigorous activity [27]. Fixation with four ISs has shown higher biomechanical strength until failure load for this UHMWPE implant [33]. While this might represent an improvement in the technique, the most important parameters remain the strength and the stiffness measured within the limit of 3 mm displacement, as this range allows the maintenance of a functional reconstruction to stabilize the stifle. An important strength until failure load associated with a displacement over 3 mm would be inefficient, as the CrCL reconstruction would be ineffective in ensuring a functional role owing to an excessive working section of the implant inside the joint. Only two tunnels with two ISs were used in the present case, resulting in a satisfactory clinical outcome. This result was similar to a previous case report in which a similar implant and fixations were used for canine caudal cruciate ligament reconstruction [37].

This highlights the importance of maintaining the dog under controlled, restricted activity during the first six weeks after surgery until bone healing and secondary stabilization are achieved [20], which complete the functional primary stabilization provided by the implant. The ISs seemed stable along the tunnels’ axes at three and six months postoperatively with the closure of both tunnels, no signs of tunnel deformation, and no signs of intolerance. Computed tomography would have helped to confirm these observations.

Implant tensioning may also influence implant slippage. An implant has a given volume determined by the size and position of its fibers. During tensioning, the fibers become parallelized, which reduces the volume of the implant’s working length (i.e., the section of the implant between the fixation points). The tension and conformation of the fibers are then maintained by the ISs. Should excessive tension be applied to the implant before fixation, its reduced volume might trigger slippage over time at the bone/IS interface as the implant would not be pressed tightly enough against the tunnel wall. Appropriate tensioning should thus provide sufficient stiffness for stifle stability while avoiding excessive tension that may initiate slippage. Implant loosening due to slippage at the bone/IS interface may explain the recurrence of mild craniocaudal translation.

Over the last decade, intra-articular reconstruction of the CrCL has shown recurrent issues of infection with allografts fixed by UHMWPE sutures with cross pins and screws with spiked washers [24], and synthetic implants made of different materials fixed by ISs and screws with spiked washers [27,28]. Mild local synovitis was observed with allografts [25] and autografts [45] fixed by UHMWPE loops and knots and with UHMWPE implants used as a replacement for or augmentation of the physiological CrCL [46]. 

In the present case, no signs of infection or aggravation of synovitis were observed during the six months post-surgery with the UHMWPE implant fixed by titanium ISs. The choice of an appropriate implant size was expected to avoid excessive friction with the cartilage and menisci, as well as any impingement on the caudal cruciate ligament, thus limiting local reaction. However, owing to the persistence of moderate joint effusion, local immune response to potential UHMWPE debris could not be excluded [58]. A cytopathological analysis of the synovial fluid would have helped confirm the absence of infection, inflammation, or immune response in the joint. 

No worsening of OA was observed up to six months postoperatively, which suggests that the stabilization provided by the implant and its fixation was satisfactory over this period. Further investigations are needed to confirm this preliminary outcome. Questions remain about the long-term integrity and functional fixation stability of the implant, with potential influence on the functional stabilization of the stifle. 

Recent studies conducted on large samples of dogs found that the mean age at CrCL rupture diagnosis was 7.1 years old [3], while dogs’ lifespans range between 4.5 and 13 years, depending on the breed [59]. The implant is expected to provide direct stabilization until the healing process is achieved with the presence of periarticular fibrosis, which contributes to stabilizing the joint [20]. Future research should investigate whether the UHMWPE implant is sufficient to provide functional stabilization for six to eight years (i.e., until the dog’s death) to avoid any aggravation of the physiological osteoarthritic degenerative process that occurs with age.

Future research should also investigate candidacy for this surgical technique. The aiming device used adapts to the morphology of each patient and allows the drilling of customized bone tunnels to fit the CrCL footprints. It would be of interest to investigate whether successful outcomes are reproducible in dogs with various tibial plateau angles, as in small-breed dogs (<15 kg) that have a steep tibial plateau angle [60]. 

A cohort study involving several centers and surgeons, more dogs, and a more thorough follow-up should provide these insights. Examinations could include computed tomography or magnetic resonance imaging to assess the tunnels, soft tissues, and periarticular healing structures; an arthroscopic evaluation to assess the implant in situ; and arthrography to observe long-term (i.e., three to four years postoperatively) development or worsening of OA. A force platform analysis might help to objectify the symmetric weight-bearing between the operated and contralateral hindlimbs [24,44].

## 5. Conclusions

In light of these preliminary results, this surgical procedure may be considered a possible treatment for CrCL rupture. This procedure, achievable under arthroscopic guidance, respects the physiological anatomy of the dog, but further investigations are needed to confirm this satisfactory outcome in one case.

## Figures and Tables

**Figure 1 vetsci-11-00334-f001:**
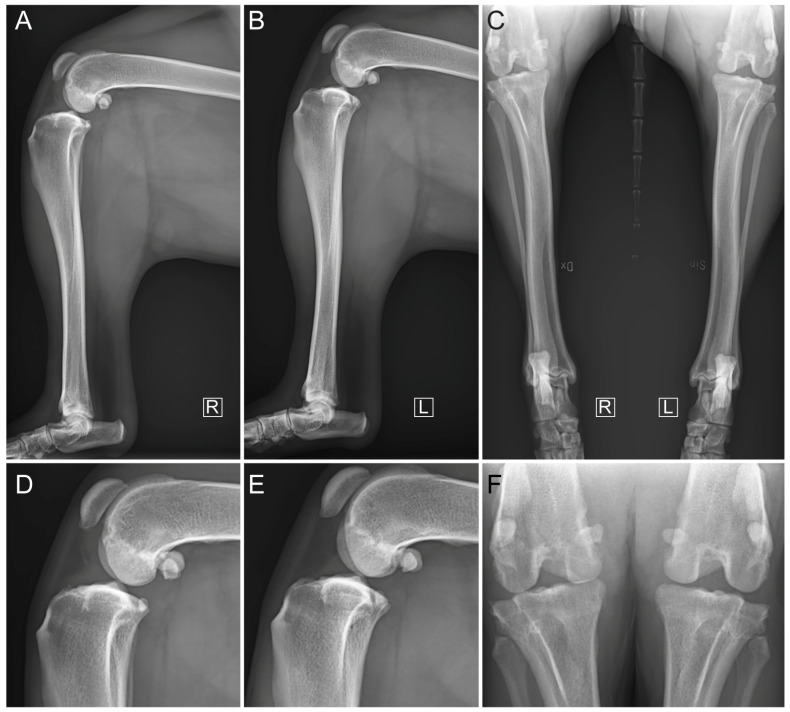
Preoperative mediolateral compression stress radiographs for (**A**) affected hindlimb and (**B**) contralateral hindlimb, and (**C**) craniocaudal radiographs for both affected and contralateral hindlimbs. Focused mediolateral view of stifle joint of (**D**) affected hindlimb and (**E**) contralateral hindlimb. Focused craniocaudal view of stifle joint of (**F**) affected hindlimb and contralateral hindlimb. R and L indicate laterality as right and left, respectively.

**Figure 2 vetsci-11-00334-f002:**
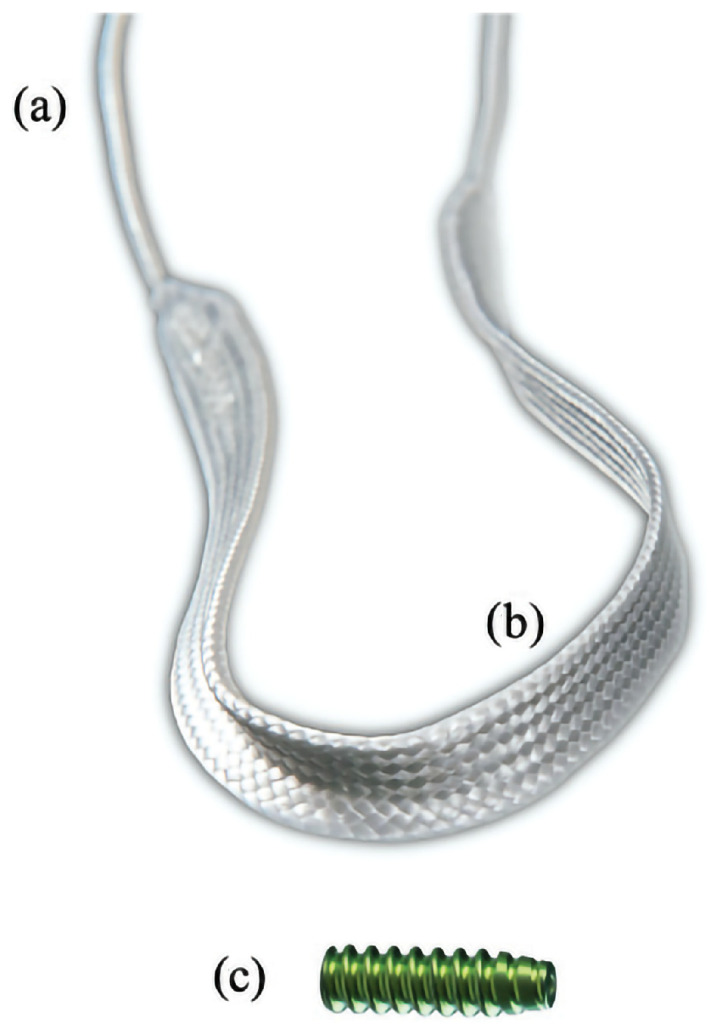
Ultra-high-molecular-weight polyethylene implant with (**a**) puller wire section and (**b**) intra-articular functional section, and (**c**) interference screw for securing implant in bone tunnels. Image used with author’s authorization [34].

**Figure 3 vetsci-11-00334-f003:**
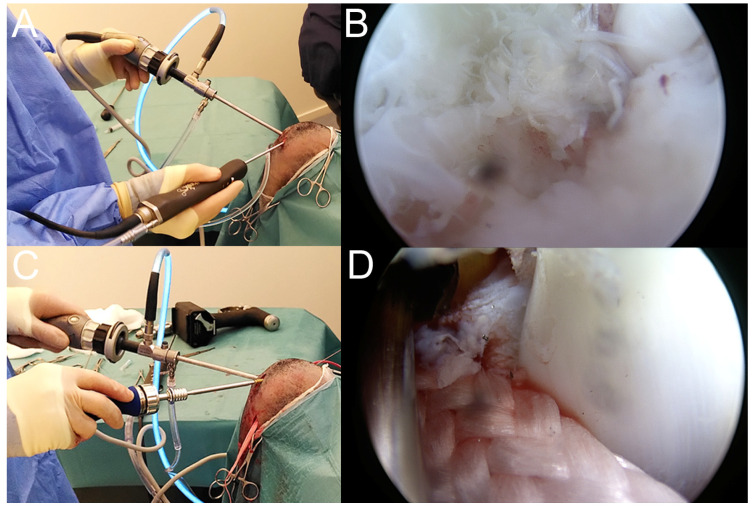
Peroperative procedures. (**A**) Introduction of arthroscope and shaver through parapatellar incisions. (**B**) Arthroscopic view of the complete ruptured cranial cruciate ligament. (**C**) Insertion of femoral interference screw. (**D**) Arthroscopic view of femoral interference screw fixing the implant.

**Figure 4 vetsci-11-00334-f004:**
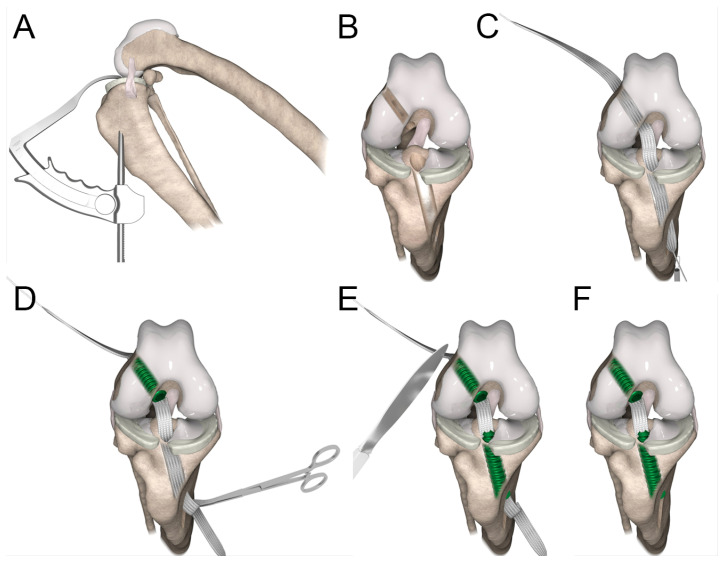
Reconstruction technique. (**A**) Medial view of stifle showing position of aiming device for drilling tibial tunnel. (**B**) Craniocaudal view of stifle showing position of femoral and tibial tunnels. (**C**) The implant was passed mediolaterally from distal to proximal through the femoral tunnel and lateromedially from proximal to distal through the tibial tunnel. (**D**) The implant was secured in the femoral tunnel by an interference screw, and tension was adjusted by rolling up the implant around Kocher forceps. (**E**) The implant was then secured in the tibial tunnel by an interference screw. Extra implant parts were removed with a scalpel (**F**).

**Figure 5 vetsci-11-00334-f005:**
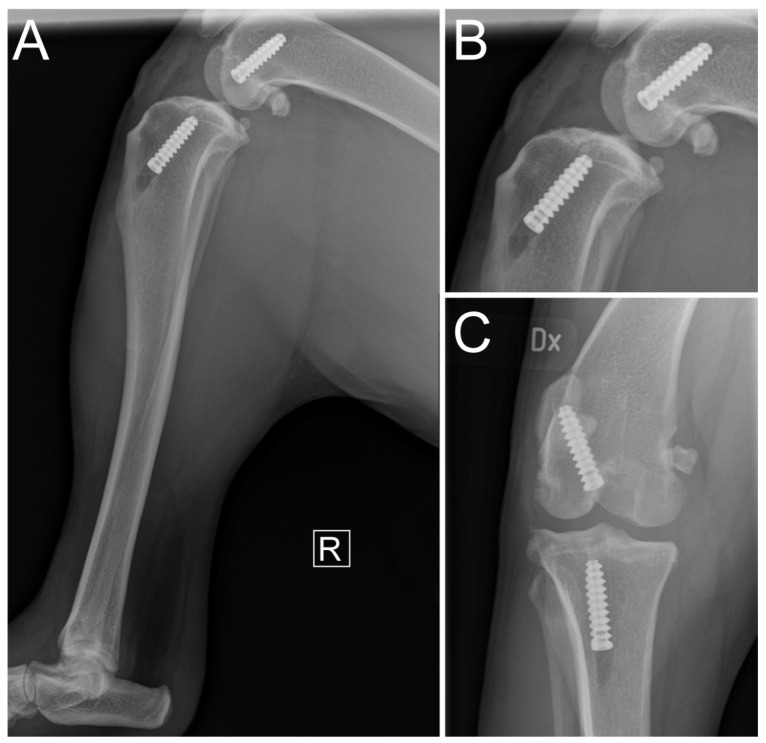
Immediate postoperative mediolateral radiographic views of affected hindlimb (**A**) with focused mediolateral (**B**) and craniocaudal (**C**) views of stifle. R indicates laterality as right.

**Figure 6 vetsci-11-00334-f006:**
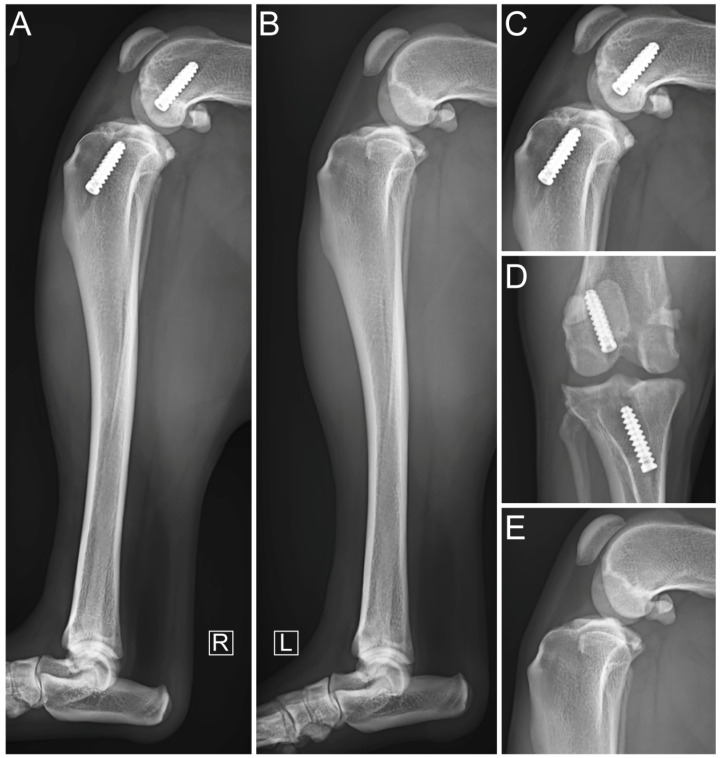
Three-month postoperative mediolateral compression stress radiographs of (**A**) operated hindlimb and (**B**) contralateral hindlimb, with focused mediolateral view of operated stifle joint (**C**) and contralateral stifle (**E**). Craniocaudal focused view of operated stifle joint (**D**). R and L indicate laterality as right and left, respectively.

**Figure 7 vetsci-11-00334-f007:**
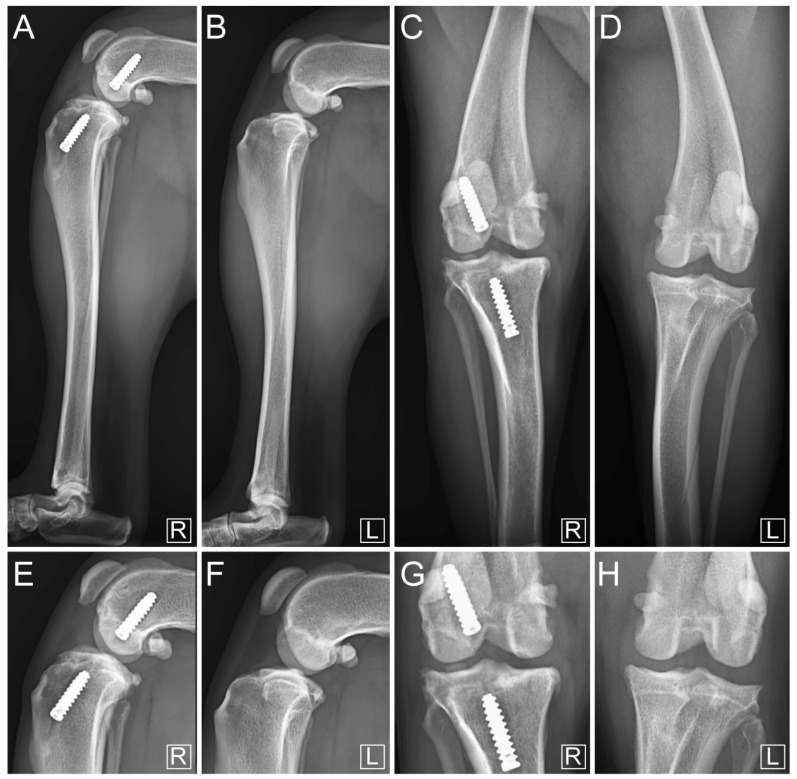
Six-month postoperative mediolateral compression stress radiographs of (**A**) operated hindlimb and (**B**) contralateral hindlimb with focused mediolateral view of operated stifle joint (**E**) and contralateral stifle (**F**). Craniocaudal view of (**C**) operated hindlimb and (**D**) contralateral hindlimb with focused craniocaudal view of operated stifle joint (**G**) and contralateral stifle (**H**). R and L indicate laterality as right and left, respectively.

## Data Availability

All the data are presented in the figures included in this article.

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
