# Peer review of "Intra-Articular Surgical Reconstruction of a Canine Cranial Cruciate Ligament Using an Ultra-High-Molecular-Weight Polyethylene Ligament: Case Report with Six-Month Clinical Outcome"

_vetsci, 2024, doi:10.3390/vetsci11080334_

Round 1

Reviewer 1 Report

Comments and Suggestions for Authors

Reviewer 1.

This manuscript add more insights about the intra-articular stabilization of the CCLR deficient stifle. The paper is overall well written. However, minor corrections are needed to clarify some points and make the understanding of the technique simpler for the reader.

Please follow the suggestions as follow:

Line 30-31: please remove “ replaces the disrupted CrCL”. It’s redundant.

Line 31: please add “allows” just before to restore.

Line 32: please add “still” before marginal

Line 34: please use “under arthroscopic guidance” all along the paper

Line 35: please change in: “the dog was weight-bearing”

Line 36: please change “few” with “minimal” or “mild” or “moderate”

Line 36: please add “post-operative” to 3 months

Line 36: please change slight with minimal or mild

Line 40: was that visible on x-rays? Please correct accordingly       

Line 56: please remove “first type of surgical management”  and add proximal to tibial osteotomies

Line 63: please remove “ “second type of surgical management”

Line 72: please rephrase this sentence: the tibial osteotomy and extra-articular technique do not address only the symptoms

Line 90: please change “during playing phase” with “after playing”

Line 92: “general examination was unremarkable”

Line 92-93: please change  in: 1/4 weight bearing lameness……

Line 95-96: that is an extrapolation of your results. Better to say the measurement ……was comparable to the contralateral one

Line 100: change in: “these findings were consistent with CrCL rupture”

Line 102: please add “compression” to stress radiographs

Line 105: please change bone proferation with “enthesophytes at patella apex”

Line 109: please remove “ and suggested an acute CrCL rupture”. It is redundtant

Line 113: please remove “ diagnostic”.

Line 132: can the authors be more specific with “ minor damage of the medial meniscus”? Which type of lesions exactly?

Line 134: at beginning the scope was medial. I imagine it was passed laterally to introduce the guide medially? Please be clearer here

Line 139: change “it” with “The tibial bone tunnel”

Line 140-144: It is not clear here from where and how the k-wire and the cannulated drill bite passed. Please be more specific here. Please describe the technique also giving more details about the exit/entry point on the tibia and femur outside of the joint.

Line 145: which kind of tap did the authors use? Please be specific

Line 146-156: please rephrase this part of the technique. It is not clear from where to where, and how the ligament is placed. Please use also the terms “proximo-distal” and “disto-proximal” or other terminology you prefer to help in the description. Keep in mind that “medio-laterally” for a lateral condyle femur is difficult to understand.

Line 156-163: how the size (length) of the screw is chosen? If on the femoral side I imagine the head of the screw is flush with the articular surface, what about the tibial side? Can the authors give more details here?

Line 206: How the drawer test was at 1month post-op?

Line 214-216: It is challenging to judge a cranial displacement of 3 mm vs 5 mm (as before the surgery). Can the authors use the compression stress radiograph as more objective toll? Is that can be done also for the pre-operative x-rays?

Line 249-256: All along the discussion, despite the authors gave all the potential technical causes to justify the residual cranial tibial translation, it is not clear if for them that is considered something expected or something not normal to avoid. If the drawer test was negative at 1 month post-op, how the technique can be improved?

Line 256: the bibliography you provide here does not show always satisfactory results on the long term, and in same articles the use of intra-articular ligament reconstruction is not advised with same implants. Please check that and change accordingly

Line 270: please change “complexifies” with “makes more complex”

Line 332: Would have a 4-tunnels approach been more appropriate for this dog, considering the size? Based on this experience, and the biomechanical ex vivo studies, are the authors changing their approach of ligament fixation?

Author Response

Comment 1: This manuscript add more insights about the intra-articular stabilization of the CCLR deficient stifle. The paper is overall well written. However, minor corrections are needed to clarify some points and make the understanding of the technique simpler for the reader.

Please follow the suggestions as follow:

Response 1: We thank the reviewer for valuable feedback, comments and suggestions that improved the clarity of our manuscript. We answered the comments in the point-by-point answer below.

Comment 2: Line 30-31: please remove “ replaces the disrupted CrCL”. It’s redundant.

Response 2: We edited the manuscript according to the reviewer’s suggestion (lines 30-31).

Comment 3: Line 31: please add “allows” just before to restore.

Response 3: We edited the manuscript according to the reviewer’s suggestion (line 31).

Comment 4: Line 32: please add “still” before marginal

Response 4: We edited the manuscript according to the reviewer’s suggestion (line 32).

Comment 5: Line 34: please use “under arthroscopic guidance” all along the paper

Response 5: We edited the manuscript according to the reviewer’s suggestion to indicate that the surgery was performed under arthroscopic guidance.

Comment 6: Line 35: please change in: “the dog was weight-bearing”

Response 6: We edited the manuscript according to the reviewer’s suggestion (line 35).

Comment 7: Line 36: please change “few” with “minimal” or “mild” or “moderate”

Response 7: We changed “few” for “mild” (line 36).

Comment 8: Line 36: please add “post-operative” to 3 months

Response 8: We added “postoperative” after “three months” (line 37) and also after “six months” (line 39).

Comment 9: Line 36: please change slight with minimal or mild

Response 9: We changed “slight” for “minimal” (line 37).

Comment 10: Line 40: was that visible on x-rays? Please correct accordingly

Response 10: These observations were visible on radiographs. We added “On radiographs,” at the beginning of the sentence to make it clear (line 41).        

Comment 11: Line 56: please remove “first type of surgical management” and add proximal to tibial osteotomies

Response 11: We edited the sentence according to the suggestion of the reviewer. The sentence is now: “Proximal tibial osteotomy modifies the biomechanical conformation of the stifle and suppresses the need for a functional CrCL during weight-bearing.” (lines 58-60).

Comment 12: Line 63: please remove “ “second type of surgical management”

Response 12: We edited the sentence according to the suggestion of the reviewer. The sentence is now: “Extra- or intra-articular stabilization of the stifle using sutures or implants are less invasive than tibial osteotomy techniques.” (lines 65-67).

Comment 13: Line 72: please rephrase this sentence: the tibial osteotomy and extra-articular technique do not address only the symptoms

Response 13: We consider the rupture of the CrCL as the direct cause of the instability symptom. Strong tibial plateau angle (TPA) is associated with CrCL rupture but is an indirect cause contributing to instability.

When the CrCL breaks, instability appears and can be corrected by modifying the TPA.

But, should the CrCL remain intact, no instability symptom would be observed even in presence of a strong TPA.

For this reason, we consider tibial osteotomy only addressing the symptom since it stabilizes the stifle by modifying the TPA, which contribute to the instability but do not cause it directly. Our logic is the same for extra-articular stabilization which only addresses the instability but do not repair the CrCL.

As this part of the sentence might be prone to interpretation while not contributing to the introduction, we deleted it. The new sentence is now “Intra-articular stabilization aims at solving the instability by reconstructing the disrupted CrCL and restoring its function [21].” (lines 74-76).

Comment 14: Line 90: please change “during playing phase” with “after playing”

Response 14: We edited the manuscript according to the reviewer’s suggestion (lines 92-93).

Comment 15: Line 92: “general examination was unremarkable”

Response 15: We changed “normal” for “unremarkable” (line 95).

Comment 16: Line 92-93: please change  in: 1/4 weight bearing lameness……

Response 16: We edited the sentence, which is now: “1/4 weight-bearing lameness of the right pelvic limb was scored at both walking and trotting [27].” (lines 95-96).

Comment 17: Line 95-96: that is an extrapolation of your results. Better to say the measurement ……was comparable to the contralateral one

Response 17: We followed the reviewer’s suggestion. The sentence is now: “Measurement of thigh muscle mass performed at 50% of thigh length was comparable to the contralateral one [40].” (lines 98-100).

Comment 18: Line 100: change in: “these findings were consistent with CrCL rupture”

Response 18: We edited the sentence, which is now: “These findings were consistent with CrCL rupture.” (line 104).

Comment 19: Line 102: please add “compression” to stress radiographs

Response 19: We added “compression” to “stress radiographs” and standardized it along the manuscript.

Comment 20: Line 105: please change bone proferation with “enthesophytes at patella apex”

Response 20: We edited the sentence, which is now: “Presence of enthesophytes at the patellar apex and on the tibial condyles was observed (Figure 1).” (lines 110-111).

Comment 21: Line 109: please remove “ and suggested an acute CrCL rupture”. It is redundtant

Response 21: We deleted the text as suggested (line 114).

Comment 22: Line 113: please remove “ diagnostic”.

Response 22: We deleted “diagnostic” as suggested (line 119).

Comment 23: Line 132: can the authors be more specific with “ minor damage of the medial meniscus”? Which type of lesions exactly?

Response 23: We completed the sentence with more information about the meniscal damage “Arthroscopic evaluation of the joint revealed a minor tear of the surface of the medial meniscus (type-4 lesion; [42]) and a complete ruptured CrCL (Figure 3B).” (lines 148-149).

Comment 24: Line 134: at beginning the scope was medial. I imagine it was passed laterally to introduce the guide medially? Please be clearer here

Response 24: We thank the reviewer for this comment. The arthroscope was actually introduced in the craniolateral incision and the shaver was introduced in the craniomedial one. We corrected it on lines 143-144. We also developed this passage to clarify the removal of the shaver and the introduction of the hook of the drilling guide: “The shaver was then removed to introduce the hook of the drilling guide through the craniomedial incision. The hook of the guide was anchored on the tibial plateau caudally to the transverse ligament of the menisci at the tibial insertion of the CrCL footprint under arthroscopic guidance (Figure 4A, 4B) [43].” (lines 152-156).

Comment 25: Line 139: change “it” with “The tibial bone tunnel”

Response 25: We edited the sentence according to the suggestion (line 163).

Comment 26: Line 140-144: It is not clear here from where and how the k-wire and the cannulated drill bite passed. Please be more specific here. Please describe the technique also giving more details about the exit/entry point on the tibia and femur outside of the joint.

Response 26: We have developed the description of the surgical technique to provide these details on lines 153-178.

Briefly, the CrCL insertion footprints were the entrance points of the tunnels inside the joint for both tunnels. The entrance point of the tunnels on the medial tibia was determined by two parameters. On the cranio-caudal axis, we aimed to drill the tunnel as caudal as possible in order to exploit the highest bone density in the caudal part of the tibia to increase fixation strength of the IS. On the proximo-distal axis, we drilled a tunnel that was long enough to fully embed the interference screw into the tunnel.

For the femoral tunnel, similarly to the tibial tunnel, once the CrCL footprint was located we angulated the k-wire to drill a tunnel long enough to fully embed the interference screw. However, this angulation was constrained by the intercondylar space, as we developed in discussion.

Comment 27: Line 145: which kind of tap did the authors use? Please be specific

Response 27: We actually compacted the tunnel using the interference screws intended for implant fixation. We corrected and developed the sentence to make it clear “Both tunnels were compacted using the 6x25-mm IS intended for implant fixation. Tibial tunnel was compacted through the medial incision on the proximal tibia, and femoral tunnel was compacted through the parapatellar craniomedial incision.” (lines 184-186).

Comment 28: Line 146-156: please rephrase this part of the technique. It is not clear from where to where, and how the ligament is placed. Please use also the terms “proximo-distal” and “disto-proximal” or other terminology you prefer to help in the description. Keep in mind that “medio-laterally” for a lateral condyle femur is difficult to understand.

Response 28: We have developed the description of the surgical technique to add these elements (lines 190-201). We named the skin incision used to introduce the implant and wire loop, as well as from where to where the implant was pulled. We also added this information regarding the insertion of the interference screw when fixing the ligament into the bone tunnels (lines 201-212).

Comment 29: Line 156-163: how the size (length) of the screw is chosen? If on the femoral side I imagine the head of the screw is flush with the articular surface, what about the tibial side? Can the authors give more details here?

Response 29: We followed the recommendation of the manufacturer regarding the choice of the implant model and IS size. The size and weight of the dog determined the model of the implant, which determined the size of the IS. Since the dog was 47 kg, an implant of 8000 N strength was required with IS between ø5 to ø7 mm and up to 25-mm long.

We added a sentence on lines 125-128, as these choices determined the size of the tunnels.

The head of the femoral IS was indeed flush with the articular surface. We added this information on line 206: “The head of the IS was flushed with the articular surface.”. It is not the case for the tibial interference screw. We added this information on lines 212-213 “The extremity of the tibial IS was not flushed with the articular surface.”.

Comment 30: Line 206: How the drawer test was at 1month post-op?

Response 30: The dog showed aggressive behaviour likely caused by stress and pain during stifle manipulation at the 1-month control. The stifle seemed stable when the drawer test was attempted, but due to the reaction of the dog, the sensitivity of the test was limited. We develop this paragraph to clarify this point on lines 252-257 : “The dog showed discomfort in standing position with occasional lifting of the paw and reacted aggressively upon manipulation of the stifle to contact with the surgical wound, which indicated a 1/3 pain score [41]. The stifle seemed stable, but because of the reaction of the dog, the sensitivity of the test was limited. No general anesthesia was induced and no additional drawer test nor manipulation of the stifle were per-formed to avoid excessive stress to the animal.”

Comment 31: Line 214-216: It is challenging to judge a cranial displacement of 3 mm vs 5 mm (as before the surgery). Can the authors use the compression stress radiograph as more objective toll? Is that can be done also for the pre-operative x-rays?

Response 31: We agree with the reviewer. We found a preprint manuscript “Quantitative evaluation of canine cranial cruciate ligament disease by stress radiography” by Kaimoto et al. 2020 (never published) where the authors used compression stress radiographs to assess and measure cranial displacement.

We tested this same approach on the present case (and on radiographs of other dogs with and without CrCL rupture) to objectify our measurement before and after surgical stabilization. We found this approach being strongly dependent on the reproducible orientation, position and pressure exerted on the limb during radiograph acquisition. We also found it as being not a good indicator of cranial displacement for all type of dogs according to tibial plateau angle or secondary stabilizers as muscle mass. For this reason, we did not use this method.

Comment 32: Line 249-256: All along the discussion, despite the authors gave all the potential technical causes to justify the residual cranial tibial translation, it is not clear if for them that is considered something expected or something not normal to avoid. If the drawer test was negative at 1 month post-op, how the technique can be improved?

Response 32: This residual non-pathological craniocaudal translation of the tibia might be expected. The instability remained in the range of values considered as stable after surgical reconstruction. Should the instability be superior to 3 mm, this could have indicated a potential issue that could influence the outcome.

In the present case, the craniocaudal translation did not influence the outcome. It was only detected with the direct drawer test. The indirect drawer test was negative. This situation is similar to the outcome obtained after TPLO where significant craniocaudal translation are still present with the direct drawer test despite stable stifle during indirect drawer test and weight-bearing.

We clarify this point on lines 312-316: “This non-pathological 3-mm craniocaudal translation, observed only with the direct drawer test, might be expected and was at the threshold value to be considered as null [24], suggesting relative joint stability. The absence of instability with the indirect drawer test, similar to weight-bearing after a TPLO [8,44], supported this satisfactory outcome.”.

Among all the parameters developed in the discussion, the insertion of the tibial interference screw flush with the articular surface was the main point of improvement to reduce the length of the implant’s working section and limit implant relaxation and thus craniocaudal translation.

We developed this idea on lines 357-360: “Inserting the IS flush with the articular surface in the tibial tunnel would reduce the length of the working section of the implant, which might help limit implant relaxation and thus craniocaudal translation.”

Comment 33: Line 256: the bibliography you provide here does not show always satisfactory results on the long term, and in same articles the use of intra-articular ligament reconstruction is not advised with same implants. Please check that and change accordingly

Response 33: We agree with the reviewer that outcomes obtained in these articles were not always satisfactory. In a given study, some dogs might have very good outcomes with no lameness and pain while other dogs show recurrent lameness, breakage of implants or infections (see Barnhart et al. 2016). We accounted for this in our sentence by mentioning that the outcome of our patient was “comparable to the best outcomes obtained” (line 308). The rest of the sentence has been edited to account for the diversity of the grafts, implants and sutures used “The satisfactory joint function of the operated stifle was similar to that of the contralateral hindlimb and comparable to the best outcomes obtained in previous intra-articular CrCL reconstructions performed with grafts [24,25,44,45], synthetic implants made of different materials [27,28] or UHMWPE sutures [31,46].” (lines 307-310).

“Synthetic implant” is a generic wording to say that an implant is made of chemical material. However, the implant we used is made of UHMWPE, a material different from the implant used in Barnhart et al. 2016 and Johnson and Conzemius 2022. The biocompatibility, the mechanical strength of the fibers and the braiding are different (see implant’s picture from Barnhart et al. 2016 and our figure 2). This means that all synthetic implants are different.

We clarified this fundamental aspect along the discussion, as poor results for a given implant/material are not representative of all types of synthetic implants.

Lines 364-367, about implant strength and integrity: “UHMWPE implants have suffered fewer breakages [31,46] than other synthetic models made of ultra-high molecular weight polyethylene terathalate and non-expanded polytetrafluoroethylene [27] or polyethylene terephthalate [28], and who were not recommended for clinical application in CrCL reconstruction.”

Lines 420-423, about infection: “Over the last decade, intra-articular reconstruction of the CrCL has shown recur-rent issues of infection with allografts fixed by UHMWPE sutures with cross pins and screws with spiked washers [24], and synthetic implants made of different materials fixed by IS and screws with spiked washers [27,28]”.

Comment 34: Line 270: please change “complexifies” with “makes more complex”

Response 34: We edited the sentence according to the suggestion (lines 326-327).

Comment 35: Line 332: Would have a 4-tunnels approach been more appropriate for this dog, considering the size? Based on this experience, and the biomechanical ex vivo studies, are the authors changing their approach of ligament fixation?

Response 35: A 4-tunnel approach might represent an improvement of the technique. However, screw number is not the only parameter to consider. Even if increasing the number of interference screws increases the strength until failure load, it is important to consider the strength and stiffness provided by the fixation within the 3-mm range of displacement, which is the limit considered as acceptable for surgical reconstruction. An important strength until failure load with an associated displacement over 3 mm would be not appropriate as the implant would be still fixed, but non-functional as the working section would be too long to stabilize the joint. In the present case, the dog has a stable stifle with only two interference screw, which was comparable to an outcome obtained with the same implant for the reconstruction of the caudal cruciate ligament.

We developed the paragraph on lines 391-403 to explain this point: “Previous synthetic reconstructions used a four-tunnel approach with IS and screws with spiked washers [27,28]. This fixation method aimed to resist a quick return to vigorous activity [27]. Fixation with four IS has shown higher biomechanical strength until failure load for this UHMWPE implant [33]. While this might represent an improvement of the technique, the most important parameters remain the strength and the stiffness measured within the limit of 3-mm displacement, as this range allows maintaining a functional reconstruction to stabilize the stifle. An important strength until failure load associated with a displacement over 3 mm would be inefficient, as the CrCL reconstruction would be ineffective in ensuring a functional role owing to an excessive working section of the implant inside the joint. Only two tunnels with two IS were used in the present case and resulted in satisfactory clinical outcome. This result was similar to a previous case report in which similar implant and fixations were used for canine caudal cruciate ligament reconstruction [37].”

Reviewer 2 Report

Comments and Suggestions for Authors

Thank you for the opportunity to review your case report. I found the manuscript to be well written and the surgical technique and follow-up to be thoroughly described. I have outlined several areas that I would like to see edited or typographical errors that require correction.  

I would also like to know what you measured for this dog’s tibial plateau angle, and to see added to the discussion what may be some considerations of candidacy for this surgical technique. Do you think TPA or other factors might play a role in the successful outcome of an intra-articular synthetic ligament reconstruction? 

Line 36: Use of “despite few signs of pain” here is confusing here in the context of the rest of the sentence. Are you trying to say that the dog’s standing posture and gait was normal at one month but continued to demonstrate signs of ongoing pain? Consider rewording here to improve clarity. 

Line 36, 38,95-96, 209, 229: The terminology “amyotrophy” is not correctly used here and would be more appropriate to be edited to “muscle atrophy”. 

Line 51: “...are more exposed to this pathology.” Awkward wording, consider editing to “are predisposed” or “are at greater risk” to this pathology.” 

Line 162: Typographical error “plan by plan” should be “plane by plane” 

Line 199: Words missing, please edit “seemed doing well.” to “seemed to be doing well.” 

Line 201: Typographical error “consisted in clinical and orthopedic...” should be “consisted of clinical and orthopedic”

Author Response

Comment 1: Thank you for the opportunity to review your case report. I found the manuscript to be well written and the surgical technique and follow-up to be thoroughly described. I have outlined several areas that I would like to see edited or typographical errors that require correction.  

Response 1: We thank the reviewer for valuable feedback, comments and suggestions that improved the clarity of our manuscript. We answered the comments in the point-by-point answer below.

Comment 2: I would also like to know what you measured for this dog’s tibial plateau angle, and to see added to the discussion what may be some considerations of candidacy for this surgical technique. Do you think TPA or other factors might play a role in the successful outcome of an intra-articular synthetic ligament reconstruction? 

Response 2: TPA in this dog was 25.8°. We added this information on line 108. Due to the unicity of the case, further discussion about candidacy is limited. The literature on intra-articular reconstruction of the CrCL is also poor regarding this pertinent aspect.

In our clinical routine, we perform intra-articular CrCL reconstruction on all type of dogs from 10 to 65 kg with satisfactory outcome. The aiming device we used adapt to each dog morphology, which allows drilling bone tunnels customized for each patient. This likely contribute to the success of the procedure as we can target the physiological CrCL footprints.

Since information on more cases treated with this surgical procedure is not available yet, we discussed this point as a perspective for future research on lines 447-451: “Future research should also investigate candidacy for this surgical technique. The aiming device used adapt to the morphology of each patient and allow drilling customized bone tunnels to fit the CrCL footprints. It would be of interest to investigate whether successful outcome is reproducible in dogs with various tibia plateau angle as in small breed dogs (< 15 kg) that have a steep tibial plateau angle [60].”

Comment 3: Line 36: Use of “despite few signs of pain” here is confusing here in the context of the rest of the sentence. Are you trying to say that the dog’s standing posture and gait was normal at one month but continued to demonstrate signs of ongoing pain? Consider rewording here to improve clarity. 

Response 3: We edited the sentence to make it clearer: “The dog was weight-bearing just after surgery and resumed normal standing posture and gait after one month, with mild signs of pain upon stifle manipulation.” (lines 35-37).

Comment 4: Line 36, 38,95-96, 209, 229: The terminology “amyotrophy” is not correctly used here and would be more appropriate to be edited to “muscle atrophy”. 

Response 4: We changed “amyotrophy” for “muscle atrophy” along the manuscript.

Comment 5: Line 51: “...are more exposed to this pathology.” Awkward wording, consider editing to “are predisposed” or “are at greater risk” “to this pathology.”

Response 5: We changed “more exposed” for “predisposed” (line 53).

Comment 6: Line 162: Typographical error “plan by plan” should be “plane by plane” 

Response 6: We corrected the sentence (line 214).

Comment 7: Line 199: Words missing, please edit “seemed doing well.” to “seemed to be doing well.” 

Response 7: We edited the sentence (line 246).

Comment 8: Line 201: Typographical error “consisted in clinical and orthopedic...” should be “consisted of clinical and orthopedic”

Response 8: We corrected the sentence (line 249).

Reviewer 3 Report

Comments and Suggestions for Authors

Dear Authors,

It is a very interesting clinical case, with a satisfactory outcome, and it uses similar materials to what it is recommended in large animals (ruminants).

I have no specific comments in terms of the form and format of the manuscript.

I have only couple minor suggestions:

As what is the actual use of this dog in particular? just companion, does he do specific training like agility or other? was the dog able to return to its normal life as before injury?

Advance imaging is lacking, CT or MRI, certainly not indispensable but could help rule out other lesions like in the meniscus for instance. Was there no access to it or it seemed as not needed? Additionally, where intraop radiographs taken to control the screw placement and insertion? how was this done?

Finally, the reference list is very extensive, I cannot tell in detail if all of them are truly justified or not as this will require to read all of these articles and point out the cited information, but I think it could be shortened.

Out of this it is a very nice work and manuscript, as these surgeries require a high level of technicity.

Author Response

Comment 1: Dear Authors,

It is a very interesting clinical case, with a satisfactory outcome, and it uses similar materials to what it is recommended in large animals (ruminants).

I have no specific comments in terms of the form and format of the manuscript.

Response 1: We thank the reviewer for feedback and valuable suggestions that we answered in the point-by-point answer below.

Comment 2: I have only couple minor suggestions:

As what is the actual use of this dog in particular? just companion, does he do specific training like agility or other? was the dog able to return to its normal life as before injury?

Response 2: We added information to clarify these two points. The dog was a companion dog “A 47-kg (body score condition: 2/5 [39]), three-year nine-month-old entire male Rottweiler used as companion dog was presented with a one-month history of lameness and abnormal standing posture of the right hindlimb.” (lines 90-92).

The dog was able to return to a normal life as before surgery. We mentioned it on lines 272-273 “The owner reported that the dog had resumed normal activity level four months after surgery.”.

Comment 3: Advance imaging is lacking, CT or MRI, certainly not indispensable but could help rule out other lesions like in the meniscus for instance. Was there no access to it or it seemed as not needed?

Response 3: In case of tibial osteotomy, CT or MRI might indeed be of interest to control soft tissue like menisci and determine first whether there is soft tissue damage to address, since it is not always necessary to open the joint capsule for this type of surgery.

The clinic that performed the surgery in the present case exclusively performs intra-articular repair to treat CrCL rupture. Beside having no access to CT or MRI, the surgeons always need to open the joint capsule to repair the CrCL, and they directly examine the joint with arthroscopy to assess damage to soft tissue before proceeding to the repair. In the case of intra-articular repair, the use of CT or MRI preoperatively is dispensable (and avoid extra costs to the owners).

However, the use of CT and MRI might be of interest during follow-up as it allows controlling soft tissue over time after CrCL repair. The benefits of the use of CT and MRI in this context is mentioned on lines 453-457.

Comment 4: Additionally, where intraop radiographs taken to control the screw placement and insertion? how was this done?

Response 4:We did not perform intraoperative radiographs. The interference screws were canulated. We used a guide k-wire to control the placement and insertion of the screws in the axes of the tunnels. The insertion of the femoral interference screw was also controlled under arthroscopic guidance. We completed the surgical technique to clarify these points for the femoral IS on lines 201-206) “A 6x25-mm cannulated titanium IS (Novetech Surgery, France) (Figure 2C) mounted on a Kirschner wire guide and on a cannulated ratchet screwdriver was inserted into the parapatellar craniomedial incision to secure the implant from inside-out in the femoral tunnel (Figure 3D, Figure 4C-D). The Kirschner wire guide and arthroscopic guidance allowed to control that the IS insertion into the tunnel follow the drilling ax-is.” and on lines 209-212 for the tibial IS “A second 6x25-mm IS (Novetech Surgery, France) (Figure 2), also mounted on a Kirschner wire guide and on a cannulated ratchet screwdriver to control insertion, was placed outside-in in the tibial tunnel using the medial incision on proximal tibia to lock the implant (Figure 4E-F).”

Comment 5: Finally, the reference list is very extensive, I cannot tell in detail if all of them are truly justified or not as this will require to read all of these articles and point out the cited information, but I think it could be shortened.

Response 5: We agree that the reference list is very extensive. The literature about treatment of CrCL rupture in dogs is rich compared to other condition. We wanted to be as exhaustive as possible to acknowledge available resources, especially regarding the intra-articular reconstruction of the CrCL that still has limited resources in comparison to osteotomies. We checked for redundant citations and shortened the reference list when possible and pertinent. The new list contains 60 references.

Comment 6: Out of this it is a very nice work and manuscript, as these surgeries require a high level of technicity.

Response 6: We thank the reviewer for acknowledging the quality of our work.